# Microstructure Evolution of 7075 Aluminum Alloy by Rotary

**DOI:** 10.3390/ma15041445

**Published:** 2022-02-15

**Authors:** Hui Cao, Yongbiao Yang, Xing Zhang, Jin Ma, Tingyan Zhang, Zhimin Zhang

**Affiliations:** College of Materials Science and Engineering, North University of China, 3 Xueyuan Road, Taiyuan 030051, China; nuccaohui@163.com (H.C.); yangyongbiao@nuc.edu.cn (Y.Y.); nucmajin@163.com (J.M.); nuczty@163.com (T.Z.); zhangzhimin@nuc.edu.cn (Z.Z.)

**Keywords:** 7075 Al alloy, rotary backward extrusion, conventional backward extrusion

## Abstract

This study proposed a rotary back extrusion (RBE) process for an open punch, which is used to produce high-performance 7075 aluminum alloy cup-shaped piece. The RBE experiment was carried out on the Gleeble-3500 testing machine at 400 °C and compared with the conventional back extrusion (CBE). The microstructure was analyzed by optical microscope, scanning electron microscope and DEFORM-3D simulation software. The results shown that compared with CBE, RBE can significantly increase the equivalent strain value and deformation uniformity of 7075 aluminum alloy cup-shape pieces. RBE deformation increases the accumulated strain of the piece, and the rotation of the die causes the piece to produce shear strain, which increases the overall strain of the cup-shape piece. The proportion of dynamic recrystallization increases, and the grain refinement was obvious. The micro-hardness value of the RBE sample is higher than that of the CBE sample, which could be the result of grain refinement strengthening. What is more, RBE and CBE have different metal flow laws.

## 1. Introduction

Aluminum and aluminum alloys have the characteristics of high specific strength, low density, strong corrosion resistance, and good molding effect. These features prompt broad application for the alloys in the aviation, aerospace, transportation, weapon industries, etc. Therefore, the innovative research and development technology of aluminum alloys, especially the forged ones, is a basic technique and has been listed as a key development target of national defense science and technology [1,2,3,4]. 7xxx-based aluminum alloy (Al-Zn-Mg-Cu) has very excellent characteristics [5]. Among various 7xxx series aluminum alloys, 7075 aluminum alloy is widely utilized for structural applications due to its high strength/density ratio and fracture toughness [6,7,8,9].

However, 7075 aluminum alloy has poor plasticity at room temperature and is difficult to deform for the manufacturing of complex parts. Therefore, hot deformation is used to increase the formability and mechanical properties of 7075 aluminum alloy under different process parameters. The hot deformation process can significantly refine the alloy grain size, and increases the comprehensive mechanical properties of the aluminum alloy [10,11,12]. Therefore, to introduce a suitable hot deformation process is significant for improving the mechanical performance of the product. Rotary back extrusion (RBE), as a new way of hot deformation process, can generate a large shear strain through one pass deformation. It can also refine the grains, and improve the uniformity of the microstructure of the cup, altering the state of stress and strain resulting in the enhanced performance [13,14]. Yu et al. [15]. proposed a new plastic deformation forming process by RBE. The RBE process is an effective plastic deformation method for producing products with enhanced workability by introducing shear strain during the extrusion process. It can change the strain path, and can also apply more equivalent strain to the material than any simple technique. Dong et al. [16]. found that in AZ80, compared with the conventional backward extrusion (CBE) process, the RBE process can significantly improve the cumulative effective strain, grain refinement and dynamic recrystallization ratio. Che et al. [17]. confirmed the ability of the RBE process to improve the grain refinement and deformation uniformity of AZ80 magnesium alloy cup-shaped parts.

However, our previous research is mainly concentrated in the formation of the microstructure and texture of the magnesium alloy RBE process, and the evolution of 7075 aluminum alloy microstructure and performance is unclear, which will limit the further development and exploration of the RBE process. In addition, 7075 aluminum alloys have good plastic toughness and processing performance, is widely used in aerospace and military sector. Therefore, this paper will study the evolution of the 7075 aluminum alloy in the RBE process, and compare the CBE, revealing the deformation behavior of the 7075 aluminum alloy RBE process and the structure property evolution mechanism.

## 2. Materials and Methods

### 2.1. Materials and Process

The material used in this study was extruded 7075 aluminum alloy. The round bar was solution treated and heat treated at 480 °C for 24 h [18], aged at 130 °C for 24 h, and wire cut after the heat treatment. After the wire cutting, a cylindrical piece with a diameter of 17 mm and a height of 21 mm was obtained (Figure 1). RBE and CBE tests were carried out in the Gleeble-3500 thermal simulator. The cup-shaped part formed after deformation at 400 °C was obtained (Figure 2). These three pictures showed the deformed cup-shaped parts seen from different sides. Since the front end of the punch was extremely thin, its resistance was relatively large. In order to prevent the high temperature from damaging the punch, the resistance wire was welded to the punch. Heat the piece and the mold to the required temperature. The experiment was divided into two groups. The CBE and rotary reverse extrusion were carried out at 400 °C. The extrusion speed was 0.1 mm/s, and the mold was moved 16 mm in total. When rotating back extrusion, the mold needed to move 2 mm first, so that the metal in the contact part of the piece and the open punch could be filled into the groove on the end of the punch, immediately after the end of the experiment. Air quenching was performed to quickly cool the deformed billet temperature to room temperature in order to obtain the microstructure immediately after the hot formation. The experiment used graphite flakes as the lubricating material. Round and rectangular graphite flakes were placed on the contact surface of the piece and the die as well as the contact surface of the open punch and the piece, respectively. This can reduce the friction between the mold and the piece during the deformation process, and ensure the smooth progress of the experiment.

### 2.2. Microstructure Characterization

Using optical microscope (OM), scanning electron microscope (SEM), and electron backscatter diffraction (EBSD), the microstructure evolution of 7075 aluminum alloy cup-shaped piece during CBE and RBE were studied. The symmetrical longitudinal section of the sample was observed. OM and SEM were prepared on Zeiss Axio-A2m microscope and Hitachi-SU5000 SEM, respectively. Before the observation, the specimen was grounded with sandpaper, mechanically polished with Al_2_O_3_ suspension, and then corroded with Keller reagent of 2.5 mL nitric acid, 1.5 mL hydrochloric acid, 1 mL hydrofluoric acid, and 95 mL distilled water [19]. The EBSD observation was carried out on an equipped Hitachi-SU5000 SEM equipped with EBSD. The scanning step was 1.0 mm. TSL-OIM analysis software was used to obtain grain size, texture, and other related microstructure features from EBSD data.

### 2.3. DEFORM-3D Simulation

DEFORM 3D-V12.0 software was used to analyze the deformation behavior of 7075 aluminum alloy during CBE and RBE processes [20,21]. First, imported the billet and die geometry model made by the 3D modeling software were imported into the DEFORM 3D system. Second, the parameters of the mold parts and pieces in the simulation process were introduced according to the experimental process. The material model was defined as an elastoplastic body, and the mold part was defined as a rigid body. The initial grid number of the relative element type was 30,000. The Coulomb friction model was used to consider the friction conditions between the piece and the mold, and the friction coefficient was 0.3. The conjugate-gradient was chosen as the solver of the simulation. The simulated process parameters were the same as the experiment. In order to prevent the piece from rotating with the convex mold, six process ribs were arranged on the inner wall of the concave mold along the longitudinal direction in order to prevent the surface of the specimen from rotating.

### 2.4. Micro Hardness Test Method

A Vickers micro hardness tester was used to measure the hardness distribution of the alloy. The test conditions were carried out with 200 gf load and 15 s holding time. Based on different locations of the deformed specimen, the hardness sampling points were selected, and the average value was taken to improve the accuracy of the experimental results. The selected area was shown in Figure 3.

## 3. Results

### 3.1. CBE and RBE Simulation Results

#### 3.1.1. Strain Field Analysis

For the preparation of 7075 aluminum alloy cup-shaped parts, a corresponding finite element model was established. Through the comparison of the two processes of CBE deformation and RBE deformation, the metal flow mode, strain distribution characteristics and the change law of forming load during the forming process of cup-shaped parts were analyzed.

Figure 4 shows the equivalent strain distribution of 7075 aluminum alloy cup-shaped pieces under CBE and RBE deformation. The equivalent strain of the inner wall of the cup-shaped piece was significantly higher than that of the outer wall, indicating that the inner wall was the main deformation area for the back extrusion deformation. In addition, the equivalent strain distribution characteristics on the outer wall of the cup-shaped piece under the two different deformation modes were completely different. Almost the entire cup-shaped part was affected by the equivalent strain in the RBE, while the CBE had only a small part, and the equivalent strain value of the RBE was much larger than that of the CBE. What is more, the RBE eliminates the deformation dead zone.

In order to further explore the equivalent strain distribution law of 7075 aluminum alloy cup-shaped parts under rotating back-extrusion deformation, according to the flow characteristics of the cup-shaped parts, three typical areas were selected: bottom zone, corner zone, and wall zone (Figure 3). The corresponding equivalent strain law between the inner and outer walls was shown in Figure 5. For the two different deformation methods, the equivalent strain values of the three regions all showed a trend of gradual decrease from the inner wall to the outer wall. However, the strain change trend at the bottom of the CBE was not the other one. It first increased and then decreased. The metal on the outside of the piece that was not in contact with the punch did not flow in the reverse direction during the deformation process, and only performed rigid translational motion along with the deformation. Therefore, the amount of strain on the inner wall of the cup-shaped piece was significantly greater than the outer wall component, resulting in a gradual decrease in the equivalent strain value from the inner wall to the outer wall. The reason for this phenomenon in conventional back extrusion was that in the conventional back extrusion deformation, the part of the billet in contact with the punch was affected by friction and the fluidity of the metal is poor, and the fluidity of the metal inside the billet was good, so the back extrusion deformation could lead to the formation of “deformation dead zones”, areas where no metal or only a small amount of metal flows. This had also led to the gradual increase of the equivalent strain value in the 0–2 mm range of the bottom area of the CBE cup-shaped part. When the equivalent strain reached its maximum value, it still showed a gradual decrease from the inside to the outside. No such phenomenon was found in the rotating back-extrusion cup-shaped parts, indicating that the rotating back-extrusion process can eliminate the deformation dead zone and promote the metal flow, which exactly corresponds to the previous equivalent strain diagram.

#### 3.1.2. Analysis of Metal Flow Trajectory

According to the equivalent strain change law of the CBE and RBE cup-shaped parts, the RBE deformation significantly increased the cumulative strain of the piece and promoted the uniformity of the overall deformation of the component. Through the analysis of the metal flow law of the cup-shaped parts, the reasons for the difference in the equivalent strain value and distribution under the two deformation modes could be better explained. Figure 6 shows the velocity vector distribution diagrams of 7075 aluminum alloy cup-shaped pieces under two deformation modes. The velocity vector can indirectly reflect the metal flow trajectory of the cup-shaped piece during the deformation process. It can be seen from the figure that the difference in the metal flow trajectory between the open punch and the piece in the conventional back extrusion and rotary back extrusion deformation was the largest. The metal flow in the bottom area of the conventional back-extrusion cup-shaped piece mainly flowed from the center of the piece to the surroundings in the radial direction, and did not change much, while the bottom metal of the rotating back-extrusion cup-shaped piece moved circularly around the center of the piece. In addition, the RBE introduces centrifugal force due to the rotation of the die, which causes the wall to become thinner. This result shows that, compared with the CBE deformation, the rotary extrusion deformation promoted the rotation of the bottom of the cup-shaped piece and the flow of metal in the contact area of the punch and the piece, thereby eliminating the “deformation dead zone” and making the bottom. The microstructure of the region became more uniform.

In order to quantitatively characterize the difference in metal flow between the two deformation methods, the point tracking function is used to analyze the metal flow law in different areas of the piece during the deformation process. Select the six characteristic points in the contact area of the punch and the piece within the diameter of the punch for point tracking analysis. P1, P2, P3 are the three points on the left side of the piece, P4, P5, P6 are the three points on the right side of the piece, P7, P8 are the two points in the middle of the piece.The distribution of the characteristic points before and after deformation is shown in Figure 7

The results show that points P1–P6 flow to the wall area after conventional back-extrusion, while points P7 and P8 only move axially with the punch. In the rotating back extrusion, the 8 points are all on the wall, and P7 and P8 are close to the bottom. This also showed that the rotating back extrusion can eliminate the dead zone of deformation, promote the flow of the bottom metal to the wall, and make the overall structure more uniform. Moreover, it can be found that the wall area is approximately 30° inclined to the original wall. Just like rowing, after the water in front is opened by the oars, it will go back to the original place again. However, due to the increased shear stress, the rotating back extrusion does not have this phenomenon.

### 3.2. The Structure of 7075 Alloy after Solid Solution

Figure 8a shows the optical microscopes (OM) of the 7075 deformed aluminum alloy after solid solution (hereinafter referred to as the original state). It shows that the crystal grains were elongated along the ED direction forming fibrous microstructure Figure 8b shows the EBSD diagram in the original state. According to the diagram, the average grain size in the original state was 106.1 μm.

### 3.3. Texture Changes during CBE and RBE

Figure 9 shows the OM microstructure of the conventional inverse extrusion cup-shaped piece at a deformation temperature of 400 °C. Combined with the metal macroscopic streamlines of the cup-shaped piece, the main typical parts of the cup-shaped piece were divided into four: dead zone (area A), isometric zone (area B), elliptical zone (area C), and corners (area D). The metal in the area B flowed to the area C, and the metal in the area C flowed to the area D in a V shape [22].

In the bottom area of the cup-shaped piece, area A was the deformation dead zone, with an average grain size of about 92.2 μm, with coarse grains. This area belonged to the deformation dead zone observed in the simulation, because the metal in this area had poor fluidity and low deformation, dynamic recrystallization was not easy to occur, so the grains were relatively coarse. The microstructure in region B was equiaxed, with a grain size of 73.2 μm, which was about 31% smaller than the initial grains, indicating that the metal flow in this region was intense and the cumulative strain increased. Area C reflected the flow of the alloy, and as the deformation progresses, the metal in the area C would flow to the area D. The average grain size of area C was 87.1 μm, which is about 17.9% smaller than the initial grains. Area C was a mixed microstructure combining original deformed grains and dynamically recrystallized grains. The deformation in the middle of the wall of the CBE cup-shaped part is low, and the dislocation density inside the crystal grain is small, and the deformation and energy storage are small, which is not conducive to the generation of dynamic recrystallization. In addition, it could be observed that only a small amount of dynamic recrystallized grains were produced along the grain boundaries of deformed grains. This was due to the accumulation of dislocations caused by grain boundaries that could hinder the movement of dislocations. The plugging dislocation could not only provide nucleation sites for the generation of dynamic recrystallization, but also cause grain boundary migration and ultimately promote the generation of new crystal grains.

Figure 10 shows the OM microstructure of the rotating reverse extrusion cup-shaped piece. From the macroscopic metal streamline, the metal in the area A flowed to the area C under the action of the rotary deformation of the die, which showed that the rotary back-extrusion deformation can promote the flow of the bottom metal. The average grain size of area A was 7.3 μm, which was reduced by 92.0% compared with conventional back extrusion. Rotating back-extrusion deformation can promote the billet to obtain more deformation, and the increase of local strain in the crystal lead to the increase of dislocation density and deformation energy storage. The increase in deformation energy storage promoted the generation of dynamic recrystallization, and, finally, eliminated the deformation dead zone of the conventional back-extrusion cup-shaped parts. The average grain size of area B was 73.1 μm, and area C was still “mixed crystal structure”. Compared with the CBE, the grains were elongated by rotating 15° counterclockwise in the RD direction on the basis of the original CBE, and the deformation was reduced by about 20.6% compared with the CBE. The average grain size of the region D was 42.7 μm, which was reduced by about 39.6% compared with the CBE. The shear strain was increased due to the rotary extrusion. As rotational extrusion increases the shear strain and deformation, the microstructure had been significantly refined, and this area was more sensitive to changes in strain. The microstructure changes showed that the rotating back extrusion deformation could significantly improve the alloy refinement ability and the proportion of dynamic recrystallization, and at the same time improved the overall deformation uniformity of the cup-shaped piece.

Figure 11 shows the EBSD map and grain boundary orientation difference map of different regions of 7075 aluminum alloy cup-shaped parts. The circles are three deformed grains, which will be mentioned later. Among them, the EBSD diagram of the cup-shaped parts showed similar results with the OM microstructure. The average grain size of the rotating reverse extrusion cup-shaped pieces in different regions was mostly smaller than that of the conventional reverse-extrusion cup-shaped pieces, indicating that the rotating reverse extrusion deformation promoted the grain refinement of the alloy. The average grain boundary disorientation at the corners of the conventional back extrusion increased from 18.3% to 22.4%. Combined with the EBSD chart, it can be seen that this was caused by the increase in the proportion of dynamic recrystallization. In the equiaxed zone, the proportion of dynamic recrystallization of aluminum alloy under the two deformation modes was not much different. Because the rotating back extrusion caused greater strain in the billet, the newly generated dynamic recrystallized grains under the action of large strain produced local strain concentration inside the newly generated dynamic recrystallized grains, and the uneven strain distribution caused dislocation plugging. Dislocations could be gradually transformed into LAGB through cross-slip and climb. Eventually, the increase in the proportion of LAGB lead to a decrease in the average grain boundary disorientation [23,24,25]. It can be seen that as the strain increases, when the coarse deformed grains transformed into fine recrystallized deformed grains, the proportion of HAGB in the alloy would increase, and the average grain boundary orientation difference increased. This led to the generation of LAGB inside the recrystallized grains, and the average grain boundary orientation difference was reduced.

For the elliptical region of the cup-shaped piece, the average grain boundary disorientation in the regional elliptical region was reduced from 17.8 μm in the conventional back extrusion to 14.9 μm in the rotary back extrusion, although the rotary back extrusion sample had a higher dynamic recrystallization ratio. However, the shear strain increased the deformation inhomogeneity inside the new dynamic recrystallization, and the increase in the degree of lattice distortion led to an increase in the dislocation density, and a large number of dislocations promoted the generation of LAGB, so the average grain boundary disorientation decreased.

In order to further explore the deformation mechanism of 7075 aluminum alloy under rotating reverse extrusion deformation, the deformed grains in the EBSD map were selected for dynamic recrystallization mechanism analysis.

Figure 12 shows the microscopic morphology of the three deformed crystal grains at the corners, elliptical area and bottom equiaxed area of the cup-shaped piece. S stands for subgrain, D stands for dynamic recrystallization, and the following numbers represent its number. For the corners, it could be observed from the EBSD diagram that the grain boundaries of the deformed grains were “serrated”, that is, the grain boundaries were bowed outwards, and there was a large amount of LAGB inside the bowed parts, which promoted the sub-grain formation, as the deformation progressed, the sub-grains gradually transformed into dynamic recrystallization grains. This phenomenon was a typical discontinuous dynamic recrystallization [26]. In addition, sub-grains and new dynamic recrystallization were observed inside the deformed grains. Their generation was different from the nucleation mechanism of the grain boundary bowing, but was dominated by the transformation of LAGB to HAGB in the grain, which was a typical CDRX features [27,28]. For the elliptical area, the deformed grains were produced by both DDRX and CDRX mechanisms, and the sub-grains and the parent grains had similar orientations. For the isometric region, it can be seen from the figure that only five grains had CDRX, and the rest were all DDRX. Through the analysis of the three parts of the cup-shaped part, it could be seen that DDRX was the main deformation mechanism of rotating reverse extrusion deformation, and the CDRX mechanism as a secondary deformation mechanism, the newly-born dynamically recrystallized grains might have different grain orientations from the parent grains, and affect the overall texture of the alloy.

### 3.4. Hardness Test

Figure 13 shows the micro-hardness values of different areas of 7075 aluminum alloy cup-shaped parts under two deformation processes. It can be seen that in different regions, the hardness of the rotating reverse extrusion cup-shaped parts is greater than the CBE hardness, indicating that the rotating back extrusion deformation can significantly increase the micro-hardness value of the alloy. The micro-hardness is mainly affected by factors such as grain size, secondary phase, dislocations density and deformation texture. A large number of grain boundaries can hinder the movement of dislocations and cause stress concentration, which ultimately leads to an increase in the micro-hardness value. From the analysis of the microstructure, it can be seen that the crystal grains are obviously refined after the rotation back extrusion deformation. This is due to the shear stress introduced by the rotating back extrusion, the metal deformation path is more complicated, and the cumulative strain is larger, so the deformed microstructure is more fine and uniform, and its fine grain strengthening effect is strong. Although the grains in the isometric zone of the conventional back extrusion are smaller than the crystal grains in the isometric zone of the rotation back extrusion, the hardness is lower than that of the rotation back extrusion. This may be due to the generation of deformation heat during deformation and the dynamic recovery of cup-shaped parts, so the grain size increases, but the proportion of dynamic recrystallization in equiaxed zone increases, and there are a large number of sub grain boundaries. A large amount of grain boundary energy increases the hardness slightly.

## 4. Conclusions

The microstructure and texture evolution of 7075 aluminum alloy in rotating back extrusion and direct back extrusion experiments at 400 °C and an extrusion speed of 0.1 mm/s were systematically studied. The main conclusions are summarized as follows:Compared with conventional back extrusion, rotary back extrusion can significantly increase the equivalent strain value and deformation range of the cup-shaped piece. The rotating back-extrusion deformation can not only drive the bottom metal of the cup-shaped piece to rotate and increase the accumulated strain of the piece, but also causes the piece to produce shear strain, which increases the overall strain of the cup-shaped piece.Rotating back extrusion deformation can significantly refine the grains, increase the proportion of dynamic recrystallization, promote metal flow, and eliminate the “deformation dead zone” in conventional back extrusion deformation, which the grain size is reduced by 92.0% compared with the CBE, but more deformation heat will also be generated in the rotating back extrusion deformation, which will cause the grains in the forming zone to recover and grow up statically.Discontinuous dynamic recrystallization is the main dynamic recrystallization mechanism of rotary back-extrusion deformation. A large number of continuous dynamic recrystallization mechanisms occur in the shear zone of the cup-shaped piece, and less in other areas. The micro hardness values of different regions of the rotating back-extrusion cup-shaped parts are all higher than those of the conventional back-extrusion samples, which is mainly caused by the strengthening of fine grains.The metal flow in the wall area of the direct back-extrusion deviates from the original wall by about 30°, while the rotating back-extrusion does not have such a phenomenon due to the increased shear stress.

## Figures and Tables

**Figure 1 materials-15-01445-f001:**
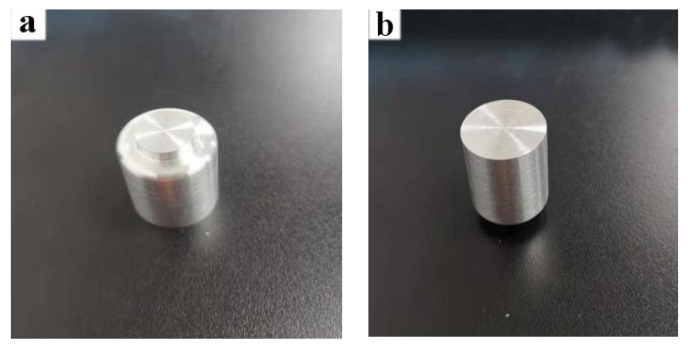
Cylindrical piece. (**a**). Top view (**b**). Bottom view.

**Figure 2 materials-15-01445-f002:**
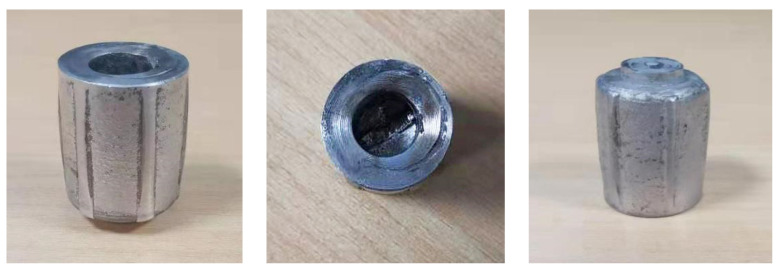
The cup-shaped piece after deformation.

**Figure 3 materials-15-01445-f003:**
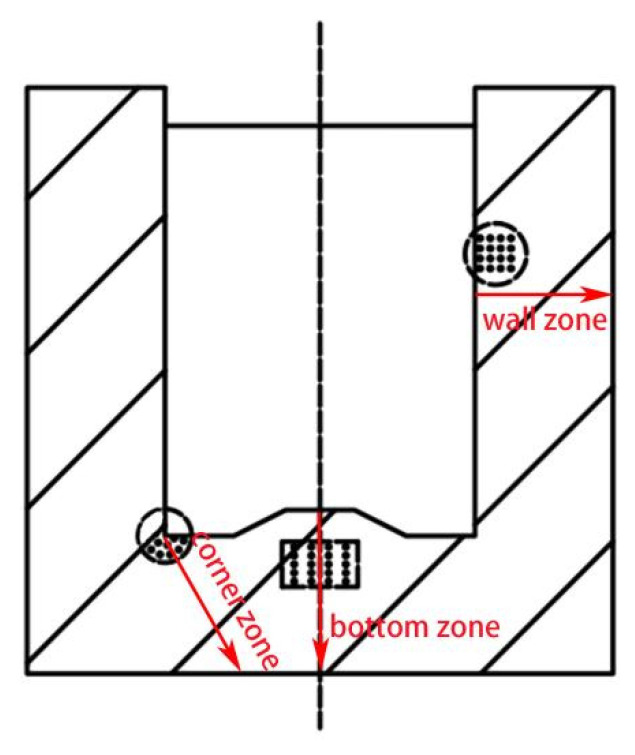
Schematic diagram of the location of the hardness test dots.

**Figure 4 materials-15-01445-f004:**
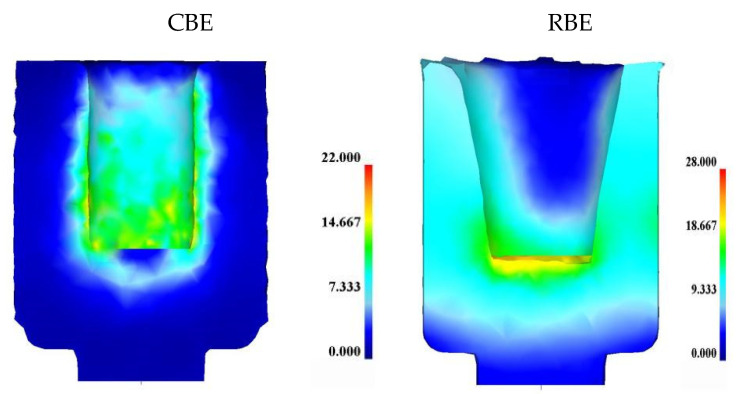
Equivalent strain distribution diagram of 7075 aluminum alloy cup-shaped piece under conventional back extrusion and rotating back extrusion.

**Figure 5 materials-15-01445-f005:**
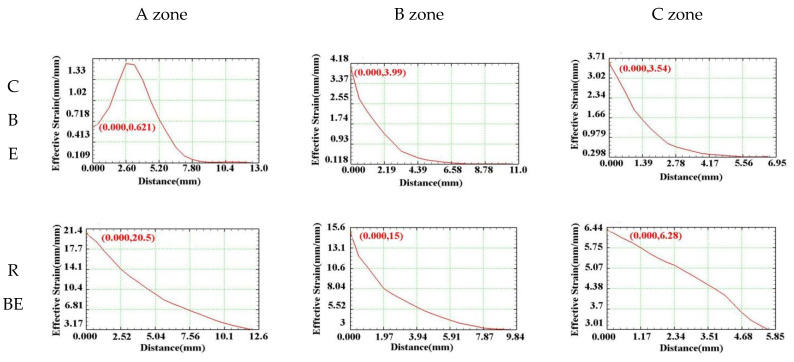
Variation law of equivalent strain between the inner and outer walls of the cup.

**Figure 6 materials-15-01445-f006:**
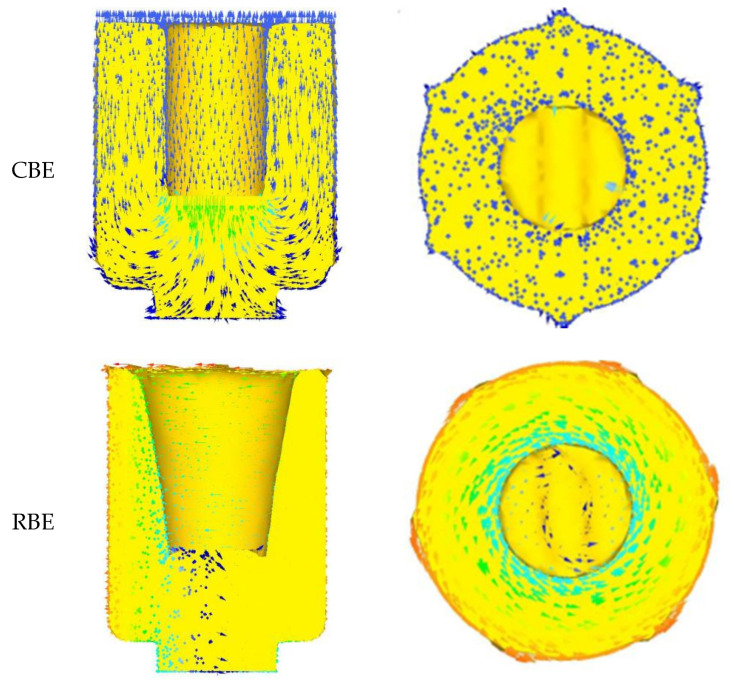
The velocity vector distribution diagram of 7075 aluminum alloy cup-shaped parts in two deformation modes.

**Figure 7 materials-15-01445-f007:**
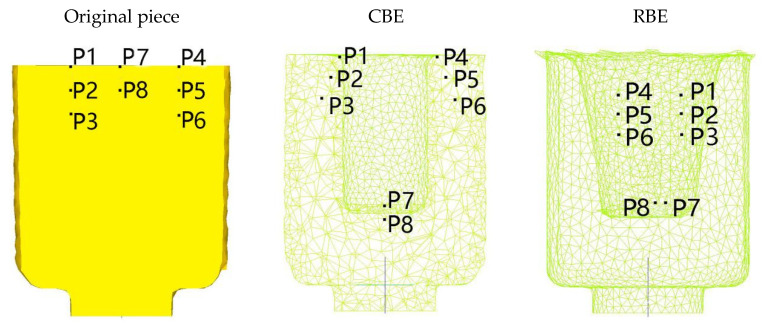
Distribution of characteristic points before and after deformation.

**Figure 8 materials-15-01445-f008:**
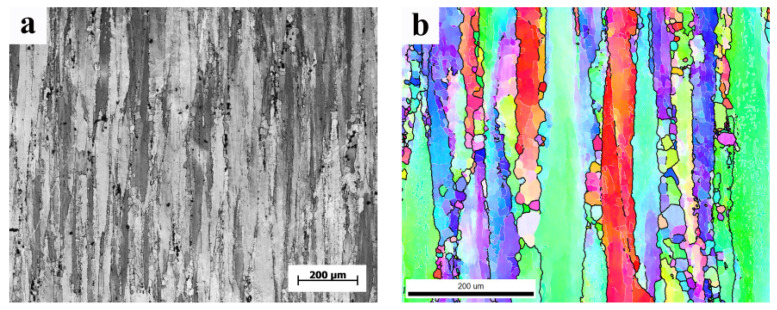
(**a**). OM of the original state; (**b**). EBSD diagram of the original state.

**Figure 9 materials-15-01445-f009:**
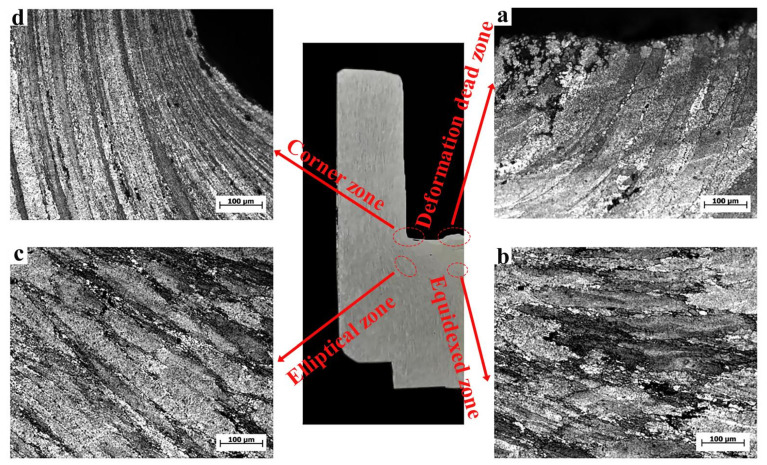
OM of CBE cup-shaped parts. (**a**) dead zone; (**b**) equidxed zone; (**c**) elliptical zone; (**d**) coener zone.

**Figure 10 materials-15-01445-f010:**
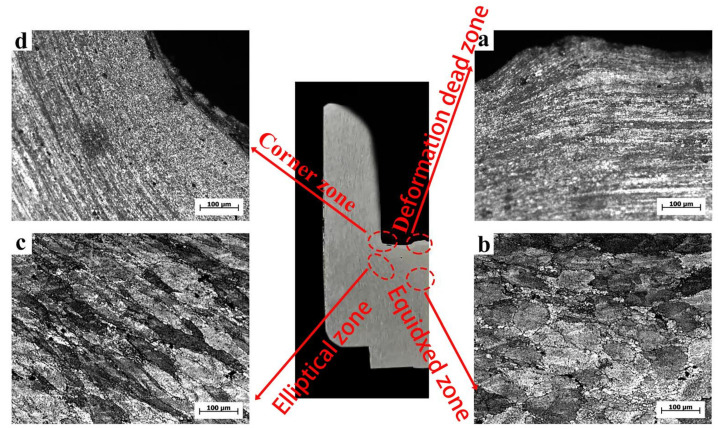
OM of the RBE cup-shaped piece. (**a**) dead zone; (**b**) equidxed zone; (**c**) elliptical zone; (**d**) coener zone.

**Figure 11 materials-15-01445-f011:**
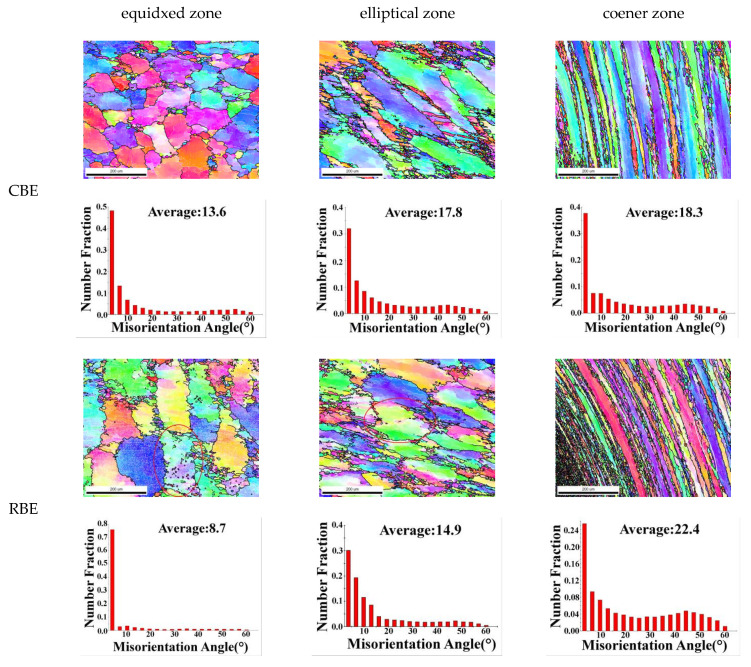
EBSD map and grain boundary orientation difference map of different regions of 7075 aluminum alloy cup-shaped piece.

**Figure 12 materials-15-01445-f012:**
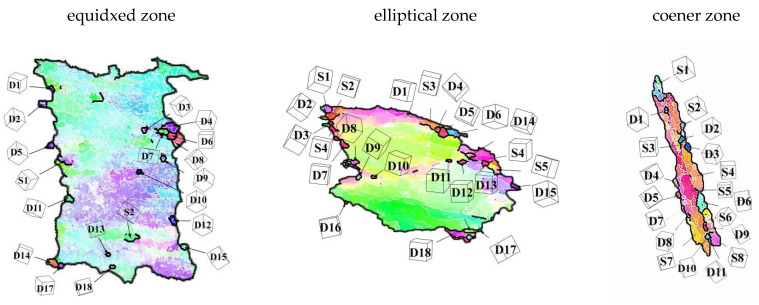
The DRX behavior of the RBE sample in Figure 11.

**Figure 13 materials-15-01445-f013:**
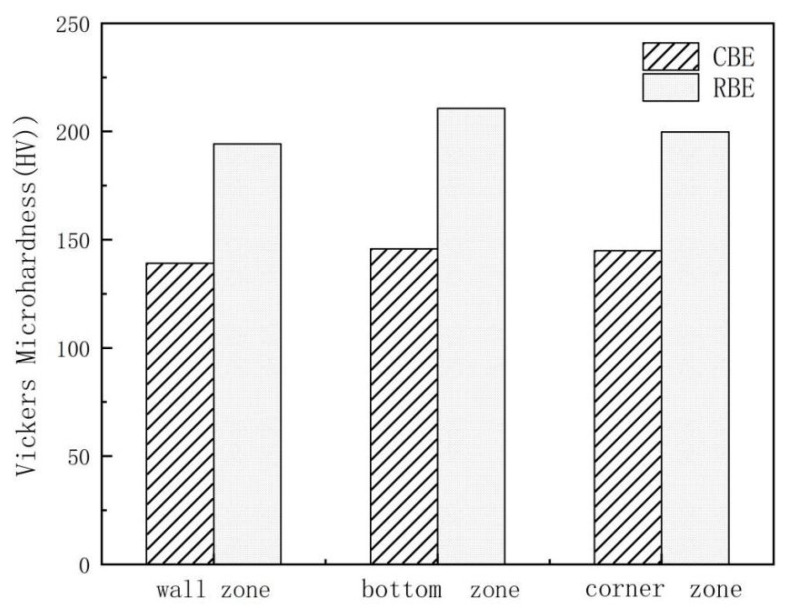
The micro hardness values of 7075 aluminum alloy cup-shaped parts in different areas under two deformation processes.

## Data Availability

Date is contained within the article.

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
