# Peer review of "Microstructure Evolution of 7075 Aluminum Alloy by Rotary"

_materials, 2022, doi:10.3390/ma15041445_

Round 1

Reviewer 1 Report

I think the paper can be accepted, but the Introduction needs small improvements.  There are very few references from the last 5 years. At the same time, the authors do not give support to the novelty of their work.

Best regardes,

Author Response

  1. I think the paper can be accepted, but the Introduction needs small improvements.  There are very few references from the last 5 years. At the same time, the authors do not give support to the novelty of their work.

According to your opinion, I have added the literature of nearly five years and revised the article.

Reviewer 2 Report

The authors have conducted a comparative analysis of the deformation behavior, structure and texture of the 7075 aluminum alloy processed by two methods, namely rotary back extrusion and conventional back extrusion, simulated using a Gleeble-3500 testing machine.

Although the paper contains some interesting results, the approaches and interpretation are not acceptable. Therefore, a very serious revision is required before the paper can be considered for publication.

The following points are critical problem for publishing.

- In this case, it is incorrect to attribute the rotary back extrusion process to the SPD method. SPD is not a technological process, but a set of physical processes in a solid [Glezer A.M., et al., General view of severe plastic deformation in solid state, Mater. Lett. 139 (2015) 455-457]. SPD includes not only ultra-high degrees of strain, but also low temperatures [R. Z. Valiev et al., Bulk Nanostructured Materials: Fundamentals and Applications, John Wiley & Sons, Inc., New Jersey, Canada, 2014]. This is usually below 0.3 of the melting point. Only for very hard-to-deform materials the temperature of the process can be slightly increased. For 7075 aluminum alloy, the temperature of 400 ºC is significantly higher than 0.3 of the melting point. Therefore, in your case, the usual hot deformation process takes place, accompanied by a dynamic recrystallization process resulting in coarse grains (although the final structure will be more dispersed compared to the original). One of the consequences of the SPD-processed aluminum alloy is the formation of a nanocrystalline structure by the low-temperature dynamic recrystallization mechanism [S.O. Rogachev et al., Effect of rotary forging on microstructure evolution and mechanical properties of aluminum alloy / copper bimetallic material, Met. Mater. Int. 2021]. Therefore, the approach should be reconsidered.

Besides a serious concern is the English in the paper. A very substantial re-editing is required before it can be considered for publication.

For example, there are unclear or grammatically/stylistically incorrect sentences:

- lines 30-31, ‘7xxx-based aluminum alloy in the aluminum alloy (Al-Zn-Mg-Cu) has very excellent …’ – unclear

- lines 68-69, ‘Heat the piece and the mold …’ - incorrect sentence form

- lines 97-98, ‘Import the billet and die geometry…’ - incorrect sentence form

- lines 109-110, ‘In the Deform-3D simulation software, the piece is set as …’ - duplication with lines 100-101

- lines 190-192, ‘Select the six characteristic points in the contact area of the punch and the piece within the diameter of the punch for …’ - incorrect sentence form

- lines 208 and 214, ‘metallographic diagram’ - incorrect term

- line 243 ,’ micro-structure diagram’ - incorrect term

- and other.

Other comments:

- some Ref. do not match the text, for example, Ref. 9-11 are far from rotary back extrusion;

- a detailed explanation is needed for Figure 2

- you need to provide a deformation scheme for the rotary back extrusion process or provide the relevant Ref.

- you need to increase the font size in figures 5 and 12; in addition, you need to re-write the caption for Figure 12

Author Response

  1. In this case, it is incorrect to attribute the rotary back extrusion process to the SPD method. SPD is not a technological process, but a set of physical processes in a solid [Glezer A.M., et al., General view of severe plastic deformation in solid state, Mater. Lett. 139 (2015) 455-457]. SPD includes not only ultra-high degrees of strain, but also low temperatures [R. Z. Valiev et al., Bulk Nanostructured Materials: Fundamentals and Applications, John Wiley & Sons, Inc., New Jersey, Canada, 2014]. This is usually below 0.3 of the melting point. Only for very hard-to-deform materials the temperature of the process can be slightly increased. For 7075 aluminum alloy, the temperature of 400 ºC is significantly higher than 0.3 of the melting point. Therefore, in your case, the usual hot deformation process takes place, accompanied by a dynamic recrystallization process resulting in coarse grains (although the final structure will be more dispersed compared to the original). One of the consequences of the SPD-processed aluminum alloy is the formation of a nanocrystalline structure by the low-temperature dynamic recrystallization mechanism [S.O. Rogachev et al., Effect of rotary forging on microstructure evolution and mechanical properties of aluminum alloy / copper bimetallic material, Met. Mater. Int. 2021]. Therefore, the approach should be reconsidered.

According to your opinion, I have attributed rotary back extrusion to thermal deformation.

  1. Besides a serious concern is the English in the paper. A very substantial re-editing is required before it can be considered for publication.

For example, there are unclear or grammatically/stylistically incorrect sentences:

- lines 30-31, ‘7xxx-based aluminum alloy in the aluminum alloy (Al-Zn-Mg-Cu) has very excellent …’ – unclear

- lines 68-69, ‘Heat the piece and the mold …’ - incorrect sentence form

- lines 97-98, ‘Import the billet and die geometry…’ - incorrect sentence form

- lines 109-110, ‘In the Deform-3D simulation software, the piece is set as …’ - duplication with lines 100-101

- lines 190-192, ‘Select the six characteristic points in the contact area of the punch and the piece within the diameter of the punch for …’ - incorrect sentence form

- lines 208 and 214, ‘metallographic diagram’ - incorrect term

- line 243 ,’ micro-structure diagram’ - incorrect term

The language in the article is improved and I revised the references.

  1. - some Ref. do not match the text, for example, Ref. 9-11 are far from rotary back extrusion;

- a detailed explanation is needed for Figure 2

- you need to provide a deformation scheme for the rotary back extrusion process or provide the relevant Ref.

- you need to increase the font size in figures 5 and 12; in addition, you need to re-write the caption for Figure 12

In addition, the explanation of the figure and the experimental scheme are added in 2.1.I improved the content of the picture and the title below.

Reviewer 3 Report

In my opinion, this paper contains serious flaws and it requires a lot of work to be published. My remarks are listed below.

  1. The langue is very odd, two examples only from one section: “crystal grains” (line 39), “(…) improve the uniformity of the micro-structure, the state of stress and strain, and the performance.” (lines 39-40), “ tissue performance evolution mechanism.” (line 57-58)!? Why “microstructure” is written with a dash? The langue has to be improved significantly!
  2. The introduction is too generic and does not explain anything. In lines 32-34 the authors wrote that the hot extrusion process can “improve the alloy structure defects” (?!) but do not even consider the fact that 7075 is a precipitation-hardened alloy and its susceptibility to temperature will result in the overaging and severe reduction of its mechanical properties during hot extrusion. Another thing is that the “hot” forming of an alloy will give us effects such as recrystallization and recovery of a grainy structure so the strain hardening will be relatively low (if any, depending on process parameters). The sentence in lines 35-37: “Severe plastic deformation of aluminum alloy is always the key to improving the strength of the aluminum alloy.” shows only that the authors do not have any knowledge or experience in aluminum alloys, have they ever heard about precipitation hardening or dispersion hardening (by zirconium or scandium)? Which aluminum alloys have higher mechanical properties – these strengthen by cold working or by precipitation hardening? The major part of the paper “research background” in this part is about a magnesium alloy AZ80.. what is the point of this? There is no research for aluminum alloys? How we can compare aluminum to magnesium? Because they are both metals..? The introduction part has to be improved!
  3. Materials and methods. What do the authors mean by “deformed 7075 aluminum alloy”? How deformed? The authors wrote that the alloy has been quenched, but there is no information if there was an artificial aging process? If so please give the parameters. Or maybe a natural aging process? If so please give me the information what was the exact time between the quenching and the performed tests. Or maybe the best way will be to just give the table containing the mechanical properties of the investigated alloy? Or its designation – T6? What was its microhardness in the initial state?
  4. Materials and methods once more. If this alloy is precipitation hardened what is the point of performing deformation at 400C (!)? There is an option to do a cold working between quenching and aging but in this case, it has literally no sense at all. In my opinion, the origin of this investigation is a severe lack of knowledge not only in terms of investigated alloy but elemental material science. Also, what does the sentence “Air quenching was performed to quickly cool the billet temperature to room temperature to avoid annealing during cooling.” mean? “Annealing during cooling”?
  5. I do not understand what is the point of the proposed microhardness measurements (Fig. 3). The authors should perform simple distribution of the microhardness on a cross-section, e.g. in zones C and A in accordance with the arrows, and refer the results to the base material’s value.
  6. The presented numerical model has no described theoretical background and lacks validation. This is unacceptable in a research paper.
  7. All values presented in this paper should have given a standard devotion. It is an overstatement that e.g. “the crystal grain size in the original state was 106.1 μm”.
  8. The EBSD study is the strongest part of this paper and presents a scientific value. In the future, it would be better to pick an alloy that can be hot deformed without overaging so this research can have an application. Nevertheless, in some processes, it has to be hot deformed e.g. in friction stir welding, as a “necessary evil” to produce a welded joint. For this reason, I think that some results from the performed investigation (EBSD predominantly) are worth publishing.

Overall: major!

Author Response

  1. The langue is very odd, two examples only from one section: “crystal grains” (line 39), “(…) improve the uniformity of the micro-structure, the state of stress and strain, and the performance.” (lines 39-40), “ tissue performance evolution mechanism.” (line 57-58)!? Why “microstructure” is written with a dash? The langue has to be improved significantly!

I revised the language of the article.

  1. The introduction is too generic and does not explain anything. In lines 32-34 the authors wrote that the hot extrusion process can “improve the alloy structure defects” (?!) but do not even consider the fact that 7075 is a precipitation-hardened alloy and its susceptibility to temperature will result in the overaging and severe reduction of its mechanical properties during hot extrusion. Another thing is that the “hot” forming of an alloy will give us effects such as recrystallization and recovery of a grainy structure so the strain hardening will be relatively low (if any, depending on process parameters). The sentence in lines 35-37: “Severe plastic deformation of aluminum alloy is always the key to improving the strength of the aluminum alloy.” shows only that the authors do not have any knowledge or experience in aluminum alloys, have they ever heard about precipitation hardening or dispersion hardening (by zirconium or scandium)? Which aluminum alloys have higher mechanical properties – these strengthen by cold working or by precipitation hardening? The major part of the paper “research background” in this part is about a magnesium alloy AZ80.. what is the point of this? There is no research for aluminum alloys? How we can compare aluminum to magnesium? Because they are both metals..? The introduction part has to be improved!

Your point of view is very correct. I improved the introduction. And there was no overaging in this experiment. This experiment is to prepare for large workpieces. We will carry out secondary solid solution and aging after deformation, so there will be no over aging problem. The background of this article is mainly to learn from the method of rotary back extrusion. There are few articles on aluminum alloy rotary back extrusion, which is also the innovation of my article.

  1. Materials and methods. What do the authors mean by “deformed 7075 aluminum alloy”? How deformed? The authors wrote that the alloy has been quenched, but there is no information if there was an artificial aging process? If so please give the parameters. Or maybe a natural aging process? If so please give me the information what was the exact time between the quenching and the performed tests. Or maybe the best way will be to just give the table containing the mechanical properties of the investigated alloy? Or its designation – T6? What was its microhardness in the initial state?

“deformed 7075 aluminum alloy” My expression is wrong. It should be extruded aluminum alloy.The piece is artificially aged. The parameter is at 2.1 in the article.The initial hardness of the piece is 137.

  1. Materials and methods once more. If this alloy is precipitation hardened what is the point of performing deformation at 400C (!)? There is an option to do a cold working between quenching and aging but in this case, it has literally no sense at all. In my opinion, the origin of this investigation is a severe lack of knowledge not only in terms of investigated alloy but elemental material science. Also, what does the sentence “Air quenching was performed to quickly cool the billet temperature to room temperature to avoid annealing during cooling.” mean? “Annealing during cooling”?

Deformation at 400 ℃ can refine the grain. And the cooling speed is fast, so there is no overaging. And there is no overaging when we make large workpieces.

  1. I do not understand what is the point of the proposed microhardness measurements (Fig. 3). The authors should perform simple distribution of the microhardness on a cross-section, e.g. in zones C and A in accordance with the arrows, and refer the results to the base material’s value.

The purpose of microhardness measurement is to measure the mechanical properties of cup-shaped parts at different positions. The hardness picture has also been improved.

  1. The presented numerical model has no described theoretical background and lacks validation. This is unacceptable in a research paper.

The simulation diagram in this paper is based on deform 3d-v12 0 software calculation, its results and the following microstructure confirm each other, and I have improved the content of 2.3.

  1. All values presented in this paper should have given a standard devotion. It is an overstatement that e.g. “the crystal grain size in the original state was 106.1 μm”.

“the crystal grain size in the original state was 106.1 μm”This is the result of EBSD post-processing.

Round 2

Reviewer 2 Report

Authors answered to most of my observations. This revised paper is now acceptable.

Author Response

Thank you for your suggestions on the revision of the article, which improves the shortcomings of the article.

Reviewer 3 Report

My remarks are listed below:

  1. In the current version of the manuscript the figures are of extremely poor quality.
  2. Answer 2: “(…) And there was no overaging in this experiment. This experiment is to prepare for large workpieces. We will carry out secondary solid solution and aging after deformation, so there will be no over aging problem.”. I do not understand what do you mean by “there was no overaging in this experiment”, but I will elaborate it in remark 5. If you plan to perform the heat treatment after deformation what is the point of pre-work precipitation hardening? It requires more force to deform it and the temperature (400C) causes a drop in mechanical properties anyway. Why did not you use 7075-O?
  3. Answer 3: “The initial hardness of the piece is 137.” In what scale? No standard deviation? This group of materials is characterized by a spread of hardness value due to large precipitates.
  4. Why it was heat treated for 48h at 480C? You are sure that the given time is correct? I also believe that there was a cooling in cold water after that.
  5. Answer 4: “Deformation at 400 ℃ can refine the grain. And the cooling speed is fast, so there is no overaging. And there is no overaging when we make large workpieces.”

- Yes, deformation at 400 C can refine grains. But when the plastic deformation is rapid (e.g. dynamic recrystallization in friction stir welding) and in your case it is 0.1 mm/s.

-  “And the cooling speed is fast, so there is no overaging”. Overaging depends predominantly on soaking temperature (400C) and time (???). What was the time of 400C affection on the material? Also, this is only one part of it, for the heating rate is unknown, cooling is the smallest problem here.

-  “And there is no overaging when we make large workpieces.” I have no idea what do you mean by this sentence. Please have a look at some aging curves of 7075 and check when the overaging occurs for given temperatures. What the size of an element has to do with it? If this temperature is needed for plastic deformation it means it affects the entire volume of the material. In aging curves there is no information about a size of an element for it is about the material only. I see another problem here – if the element is large, the heating rate has to be lower (slightly, for it is aluminum after all) and for that, an affection of temperature on the material is larger.

  1. Answer 7: ““the crystal grain size in the original state was 106.1 μm”This is the result of EBSD post-processing.”. Even if we consider that EBSD is 100% accurate and there is no standard deviation I understand that if I move the sample by 50 um and I repeat the measurement the result will be exactly the same?

If I may ask, in the next answers please add some reference/diagrams/curves for answering my remarks about the overaging problem.

Overall: major.

Author Response

  1. In the current version of the manuscript the figures are of extremely poor quality.

In the current version of the manuscript, the figures have been revised.

  1. Answer 2: “(…) And there was no overaging in this experiment. This experiment is to prepare for large workpieces. We will carry out secondary solid solution and aging after deformation, so there will be no over aging problem.”. I do not understand what do you mean by “there was no overaging in this experiment”, but I will elaborate it in remark 5. If you plan to perform the heat treatment after deformation what is the point of pre-work precipitation hardening? It requires more force to deform it and the temperature (400C) causes a drop in mechanical properties anyway. Why did not you use 7075-O?

You are right. It requires more force to deform it and the temperature (400C) causes a drop in mechanical properties anyway. But the grains of as cast 7075 aluminum alloy are larger, while the grains of extruded aluminum alloy are smaller after deformation, and dynamic recrystallization is easier to occur.

  1. Answer 3: “The initial hardness of the piece is 137.” In what scale? No standard deviation? This group of materials is characterized by a spread of hardness value due to large precipitates.

The hardness is measured before thermal deformation.

  1. Why it was heat treated for 48h at 480C? You are sure that the given time is correct? I also believe that there was a cooling in cold water after that.

I heat treated it at 480℃ for 24 hours and it was a cooling in cold water after that.

  1. Answer 4: “Deformation at 400 ℃ can refine the grain. And the cooling speed is fast, so there is no overaging. And there is no overaging when we make large workpieces.”

- Yes, deformation at 400 C can refine grains. But when the plastic deformation is rapid (e.g. dynamic recrystallization in friction stir welding) and in your case it is 0.1 mm/s.

-  “And the cooling speed is fast, so there is no overaging”. Overaging depends predominantly on soaking temperature (400C) and time (???). What was the time of 400C affection on the material? Also, this is only one part of it, for the heating rate is unknown, cooling is the smallest problem here.

-  “And there is no overaging when we make large workpieces.” I have no idea what do you mean by this sentence. Please have a look at some aging curves of 7075 and check when the overaging occurs for given temperatures. What the size of an element has to do with it? If this temperature is needed for plastic deformation it means it affects the entire volume of the material. In aging curves there is no information about a size of an element for it is about the material only. I see another problem here – if the element is large, the heating rate has to be lower (slightly, for it is aluminum after all) and for that, an affection of temperature on the material is larger.

400℃ is the temperature during deformation, and its heating speed is 10℃/s. 

Since the second phase is not very large and no overaging problem is found in the observation of optical microscope and EBSD, I did not consider the effect of over aging on grain refinement. I will continue to study it next.

  1. Answer 7: ““the crystal grain size in the original state was 106.1 μm”This is the result of EBSD post-processing.”. Even if we consider that EBSD is 100% accurate and there is no standard deviation I understand that if I move the sample by 50 um and I repeat the measurement the result will be exactly the same?

You are right. If you move the sample, its grain size will be different, but its grain size is still very large. This is the grain morphology and grain size obtained after I moved the position.

Please see the additional picture in attached file.    

Round 3

Reviewer 3 Report

Thank you for your answers. I recommend the publication.